# Single-cell transcriptome profiling and the use of AID deficient mice reveal that B cell activation combined with antibody class switch recombination and somatic hypermutation do not benefit the control of experimental trypanosomosis

Hang Thi Thu Nguyen[1,2,3], Robin B. Guevarra[2], Stefan Magez[1,2,3]*,
Magdalena Radwanska[2,4]*

1 Department of Biochemistry and Microbiology, Ghent University, Ghent, Belgium, 2 Laboratory for Biomedical Research, Ghent University Global Campus, Incheon, South Korea, 3 Laboratory of Cellular and Molecular Immunology, Vrije Universiteit Brussel, Brussels, Belgium, 4 Department of Biomedical Molecular Biology, Ghent University, Ghent, Belgium

☯ These authors contributed equally to this work.
* Stefan.Magez@ghent.ac.kr (SM); magdalena.radwanska@ghent.ac.kr (MR)

**Data Availability Statement:** scRNA-seq datasets are publicly available in the ArrayExpress database

## Abstract

Salivarian trypanosomes are extracellular protozoan parasites causing infections in a wide range of mammalian hosts, with *Trypanosoma evansi* having the widest geographic distribution, reaching territories far outside Africa and occasionally even Europe. Besides causing the animal diseases, *T. evansi* can cause atypical Human Trypanosomosis. The success of this parasite is attributed to its capacity to evade and disable the mammalian defense response. To unravel the latter, we applied here for the first time a scRNA-seq analysis on splenocytes from trypanosome infected mice, at two time points during infection, i.e. just after control of the first parasitemia peak (day 14) and a late chronic time point during infection (day 42). This analysis was combined with flow cytometry and ELISA, revealing that *T. evansi* induces prompt activation of splenic IgM+CD1d+ Marginal Zone and IgM^Int IgD+ Follicular B cells, coinciding with an increase in plasma IgG2c Ab levels. Despite the absence of follicles, a rapid accumulation of *Aicda*+ GC-like B cells followed first parasitemia peak clearance, accompanied by the occurrence of *Xbp1*+ expressing CD138+ plasma B cells and *Tbx21*+ atypical CD11c+ memory B cells. Ablation of immature CD93+ bone marrow and *Vpreb3*+*Ly6d*+*Ighm*+ expressing transitional spleen B cells prevented mature peripheral B cell replenishment. Interestingly, AID^-/- mice that lack the capacity to mount anti-parasite IgG responses, exhibited a superior defense level against *T. evansi* infections. Here, elevated natural IgMs were able to exert *in vivo* and *in vitro* trypanocidal activity. Hence, we conclude that in immune competent mice, trypanosomosis associated B cell activation and switched IgG production is rapidly induced by *T. evansi*, facilitating an escape from the detrimental natural IgM killing activity, and resulting in increased host susceptibility. This unique

under the accession number E-MTAB-10174. All other relevant data are within the manuscript and its Supporting information files.

**Funding:** This study was supported by Ghent University Global Campus core funding for MR, Ghent University Global Campus core funding for SM, UGent BOF grant number BOF. STG.2018.0009.01/01N01518 for SM and FWO grant number G013518N for SM. The funders had no role in study design, data collection and analysis, decision to publish, or preparation of the manuscript.

**Competing interests:** The authors have declared that no competing interests exist.

role of IgM and its anti-trypanosome activity are discussed in the context of the dilemma this causes for the future development of anti-trypanosome vaccines.

## Author summary

*Trypanosoma evansi* parasites can infect mammals, occasionally also humans, by evading the humoral immune response. In this study, cellular and transcriptomic profiling reveals that *T. evansi* induces rapid activation of mature splenic B cells, followed by differentiation into *Aicda*[+] GC-like B cells, *Tbx21*[+] atypical memory B cells and *Sdc1*[+]*Xbp1*[+] plasma B cells. The process triggers early-stage *Ighg2c* expression in Follicular B cells. Simultaneous ablation of the bone marrow early B cell lineage prevents B cell replenishment, causing loss of the host's parasitemia control capacity. Surprisingly, AID[-/-] mice lacking anti-parasite IgGs, exhibit a superior defense level against *T. evansi* infections, with elevated natural IgMs being able to exert trypanocidal activity. Hence, we conclude that in immune competent mice, trypanosomosis associated B cell *Aicda* activation and IgG2c production is rapidly induced by *T. evansi* in order to evade natural IgM mediated killing, resulting in increased host susceptibility.

## Introduction

Salivarian trypanosomes are single-cell extracellular protozoan parasites that reside in the blood, lymphatics and tissues of a wide range of mammalian hosts. *Trypanosoma evansi* (*T. evansi*) is classified as an animal trypanosome [1,2], but also cause atypical Human Trypanosomosis (aHT) [3–5]. This parasite has reached a near-global geographic distribution including Europe, where infections resulting from the import of parasite harboring animals have been reported [6,7]. Transmission of this parasite does not depend on the bite of a *tsetse* but can be mediated by range of different vectors [8]. The success of *T. evansi* in colonizing a broad range of hosts, including humans, infers that the parasite acquired adaptations to circumvent multiple immune killing mechanisms. As for *T. brucei*, antigenic variation of the Variant Surface Glycoprotein (VSG) coat is considered to be the first line of *T. evansi* defense against elimination by host antibodies [9]. This process involves chromosomal recombination, expression side activation/silencing and access to an extensive reservoir of over 1000 genomic VSG genes and pseudogenes, allowing trypanosomes to 'eternally' outrun the mammalian adaptive immune system [10]. However, VSG switching is not the only mechanism involved in parasite evasion of the host immunoglobulin response. Rapid lateral surface movement followed by endocytosis of antibody-complexed VSG molecules allows the clearance of surface bound Igs, and reduces the efficacy of complement mediated clearance after C3b surface deposition [11,12]. VSG shedding further decreases C3b binding [13]. Interestingly, several experimental settings involving gene-deficient mice indicate that peak control of parasite numbers does not involve active lysis by the C9 complement complex itself (reviewed by [14]. Trypanosome infections were also shown to reduce C1, C1q and C3 serum concentrations [15,16]. Finally, trypanosome infections are characterized by a severely compromised host B cell compartment [17–22]. Here, the cytotoxic effect of NK cells and contact-dependent interactions between B cells and trypanosomes were shown to induce apoptosis of Transitional B cells after upregulation of CD95 surface expression and caspase-3 activation. For *T. evansi*, the in-depth understanding of host-parasite interactions remains largely absent, even as infections in cattle, water

buffalos and pigs are known to cause the abrogation of B cell memory. The latter results in infection-associated failures of vaccines for classic swine fever [23], Foot and Mouth disease (7), *P. multocida* [24] and *P. haemolytica* [25], and the overall susceptibility to secondary infections [26]. No commercial vaccine against trypanosomosis caused by *T. evansi*, or any other trypanosome species, is available so far.

Besides multiple reports describing cellular aspects of this B cell depletion, trypanosomosis is also characterized by polyclonal B cell activation. High serum titers of anti-trypanosome antibodies as well as cross-reactive polyclonal antibodies were shown to perpetuate throughout infection [27–29]. So far, no mechanistic data has reconciled both observations simultaneously. To address this, we adopted an experimental infection model of *T. evansi*, using a non-cloned field stabilates that causes a reproducible chronic infection in mice. We deliberately opted for this approach, as most reliable field data on the failure of the host adaptive immune system comes from clinical *T. evansi* reports and vaccine failure reports in *T. evansi* infected livestock.

Our study provides new insights into the interplay between trypanosomes and the host splenic B cell compartment, identifying mechanisms involved in the modulation of the immune response. Transcriptomic analysis by scRNA-seq, flow cytometry, histology, ELISA and the use of knock out mice, demonstrates the rapid and terminal differentiation of B220$^+$CD21$^{high}$CD1d$^+$IgM$^+$ Marginal Zone B cells (MZBs), and B220$^+$CD23$^{high}$IgD$^+$IgM$^{Int}$ Follicular B cells (FoBs) into IgM and IgG2c secreting plasma cells (PCs), independently from losing splenic B follicles during infection. This coincides with ablation of the early B lineage CD93$^+$ bone marrow compartment, resulting in the absence of mature B cells replenishment in the periphery and the long-term inability of the host to control the infection. Finally, the use of AID$^{-/-}$ mice, characterized by the absence of class switch recombination (CSR) and antibody affinity maturation by somatic hypermutation (SHM), showed a superior IgM dependent resistance to *T. evansi* infection. These findings are discussed in the context of implications for anti-trypanosome vaccine development in relation to induction of long lasting protective IgM responses.

## Results

### 1. *T. evansi* Merzouga 93 causes chronic infection in mice accompanied by sustained high antibody responses

*T. evansi* Merzouga 93 causes a chronic infection in C75BL/6 mice, lasting up to 16 weeks. Despite being a non-cloned parasite, *T. evansi* Merzouga 93 infections are consistently characterized by a relatively well controlled first wave of parasitemia in the second week of infection, followed by a much higher peak around 20 days post infection (dpi) and subsequent parasitemia waves that can reach 2.8x10$^9$ parasites/ml blood (Fig 1A). Mice started to succumb to the *T. evansi* from 42dpi onward (Fig 1B). A rapid increase in total infection induced IgM antibody titer was measurable as early as 7dpi, followed by an IgG2c induction (Fig 1C). The anti-VSG Ab secretion profile followed a similar pattern, but significantly increased titers were only observed after the first parasitemia peak (Fig 1D). Infection-induced titers for total IgG2b, IgG3 and IgG1 were only detected 4–6 weeks into infection, lacking consistent VSG binding activity (S1 Fig). *T. evansi* infections result in a very significant decline in cell numbers of B220$^+$CD1d$^+$ MZBs starting on day 14 post infection, and B220$^+$CD1d$^-$ FoBs at the late stage (Fig 1E). CD138$^+$ PCs number increased rapidly, peaking at 14dpi followed by a decline in cell numbers thereafter (Fig 1E). Trypanosome infections in mice induce severe splenomegaly, mainly resulting from erythro-granulopoiesis [30]. This pathology was also confirmed in case of *T. evansi* infection (S2 Fig). MZB and FoB spleen cell identification by flow cytometry

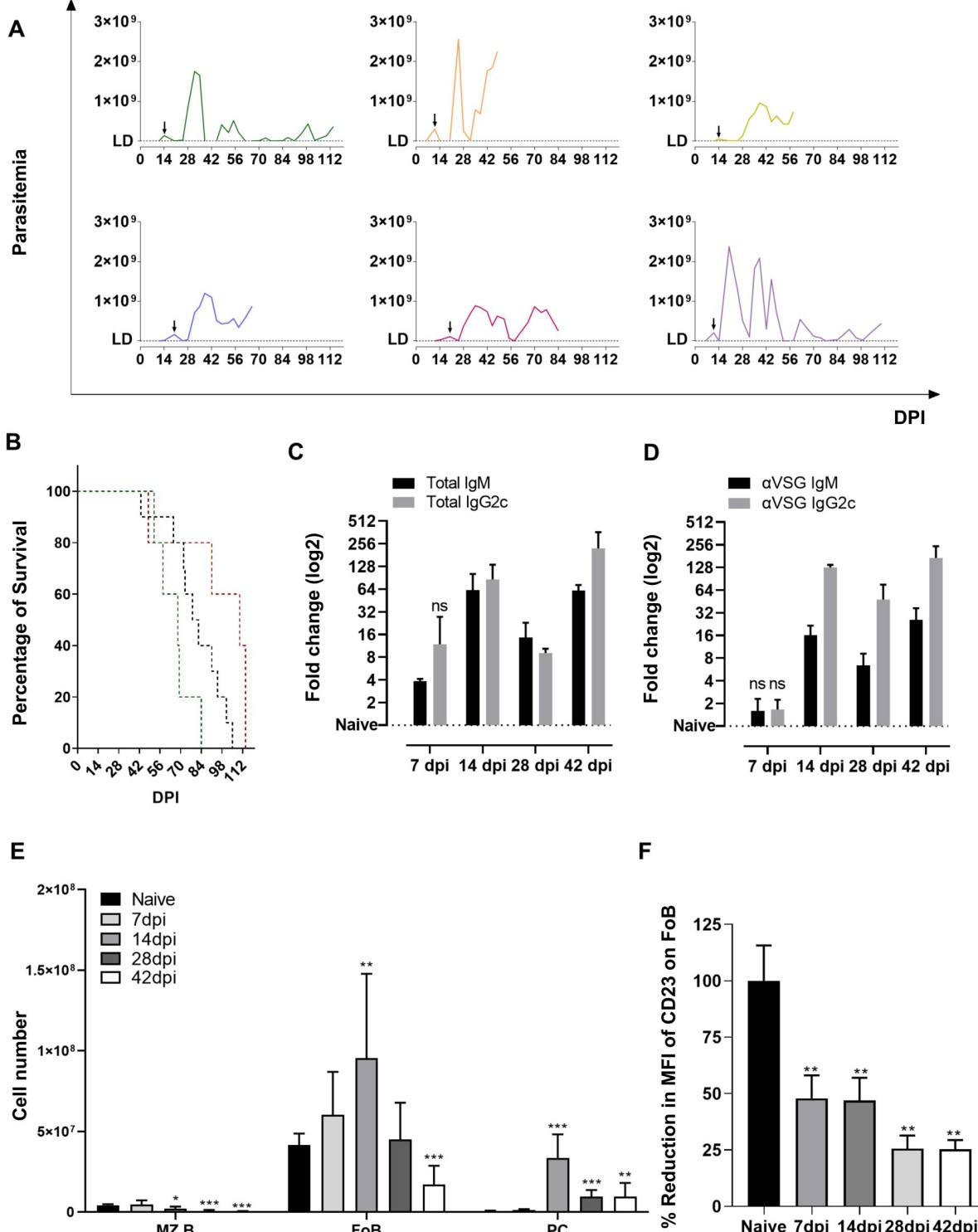

**Fig 1. *T. evansi* causes chronic infections with sustained high antibody response and loss of B cell homeostasis. (A)** Parasitemia of 6 representative infected mice, arrow indicating day of first parasitemia peak. DL: detection limit (2x10⁶ parasites/ml). **(B)** Survival curve in 3 independent experiments. Green line: n = 5, Black line: n = 10, Red line: n = 5. **(C)** Total IgM and IgG2c isotype plasma antibody titers. Data presented as log2 fold change of OD50 values of infected versus naïve samples. Results represent the means + SD of three individual samples, form one of three representative experiments. **(D)** Anti-VSG IgM and IgG2c isotype plasma antibody titers. Data presented as log2 fold change of end-point titers of infected versus naïve samples. Results represent the means + SD of three individual samples, form one of three representative experiments. **(E)** Absolute cell number of of MZBs, FoBs, and PCs present in spleen during infection. Results represent the

means + SD of 9 combined samples, obtained from 3 independent experiments. (F) Mean fluorescent intensity reduction in percentage of CD23 expression on FoBs. Results represent the means + SD of three individual samples representing one of three representative experiments, following gating strategy which is shown in S6C Fig. (C-D) Significant differences versus control (Naïve) reported as p≤0.05 by Student's t-test, ns: non-significant. (E-F) Significant reported as * p≤0.05; ** p≤0.01; *** p≤ 0.001 by Student's t-test, versus naïve values.

requires particular attention during trypanosome infections. Conventionally, CD23 is used here as discriminatory marker. However, *T. evansi* infections induce a rapid reduction of CD23 surface expression on FoBs, making it unsuitable for correct cell identification or cell gating strategies. This downregulation is already significant at 7dpi, and continues throughout infection (Fig 1F).

## 2. ScRNA sequencing analysis reveals *T. evansi* induced terminal B cell differentiation

Single cell RNA-seq analysis was performed using splenocytes from non-infected and infected mice at 14dpi (control after the 1st peak of infection) and 42dpi (time point of 1st infection-induced death in all experimental repeats combined). Unbiased graph-based clustering on the 3-time-point-aggregated dataset resulted in 35 discrete clusters in which B cells could be annotated (S3A and S3B Fig). However, due to the heterogeneity of the splenocytes samples, neither the MZB nor FoB, the 2 major mature B cell sub-populations, were well distinguishable using conventional markers (S3C Fig). To enhance in-depth exploration of these populations, B cells and PCs were extracted from individual sample and integrated to create a combined dataset. This approach yields distinct B cell lineage populations including Transitional B cells, MZB, FoB, GC-like B and PC populations, as well as a smaller heterogeneous population comprising both B1 B cells, and so-called atypical short lived memory B cells (atMBCs) (Fig 2A). The main markers used for annotation include the B lymphocyte antigen CD19, the early B cell marker CD93, the immature B cell surface molecule Ly6d, the B cell maintenance transcription factor EBF1 and the CD20 coding gene *Ms4a1* (Fig 2B). Mature B cell populations were identified using *Cr2* (CD21/35) and *Fcer2a* (CD23) in combination with *Ighd* (IgD heavy chain) while *Itgam* (CD11b), the transcriptional repressors genes upregulated during B cell activation *Zbtb32* and differentiation marker *Zbtb20* were used to annotate atMBCs and B1 B cells respectively. The *Aicda* gene and *Mki67* proliferation marker were used for GC-like B cell identification. PCs were annotated based on the presence of transcripts for immunoglobulin heavy chains in combination with *Sdc1* (CD138). A heatmap combining the top 10 signature genes of each sub-population shows very distinct B cell populations (Fig 2C). Key signature genes included *Vpreb3*, encoding a μ heavy chain associated protein present during BCR bio-synthesis and *Ly6d* for Transitional B cells. Interestingly, the mixed B1/atMBC population seen in Fig 2A contains two distinct clusters with *Zbtb20* clearly upregulated in the B1 population and *Zbtb32* being upregulated in the atMBCs. Furthermore, *Cr2* (CD21), the *Ighd* (IgD) and *Fcer2a* (CD23) differentiate MZBs and FoBs, while the proliferation marker *Pclaf* and cell cycle progression linked marker *Ube2c* marks GC-like B cells. Finally, PCs are characterized by the specific genes *Xpb1*, *Ighm* and the immunoglobulin J chain. The entire list of differentially expressed genes is represented in the S4 Fig. Based on cell type annotation, alterations of each sub-population were quantified, confirming the flow cytometry data and showing a very drastic infection-induced reduction in the splenic B cell fraction (Fig 2D). Cell numbers were subsequently calculated showing that the combined B cell lineage population initially representing nearly 34% of splenocytes in the non-infected sample, dropped to less than 1% at 42dpi (Fig 2E pie charts). Coinciding with the decrease of mature B cells, there was a striking nearly 20-fold increase in PCs at 14dpi, followed by a virtually complete loss of this population

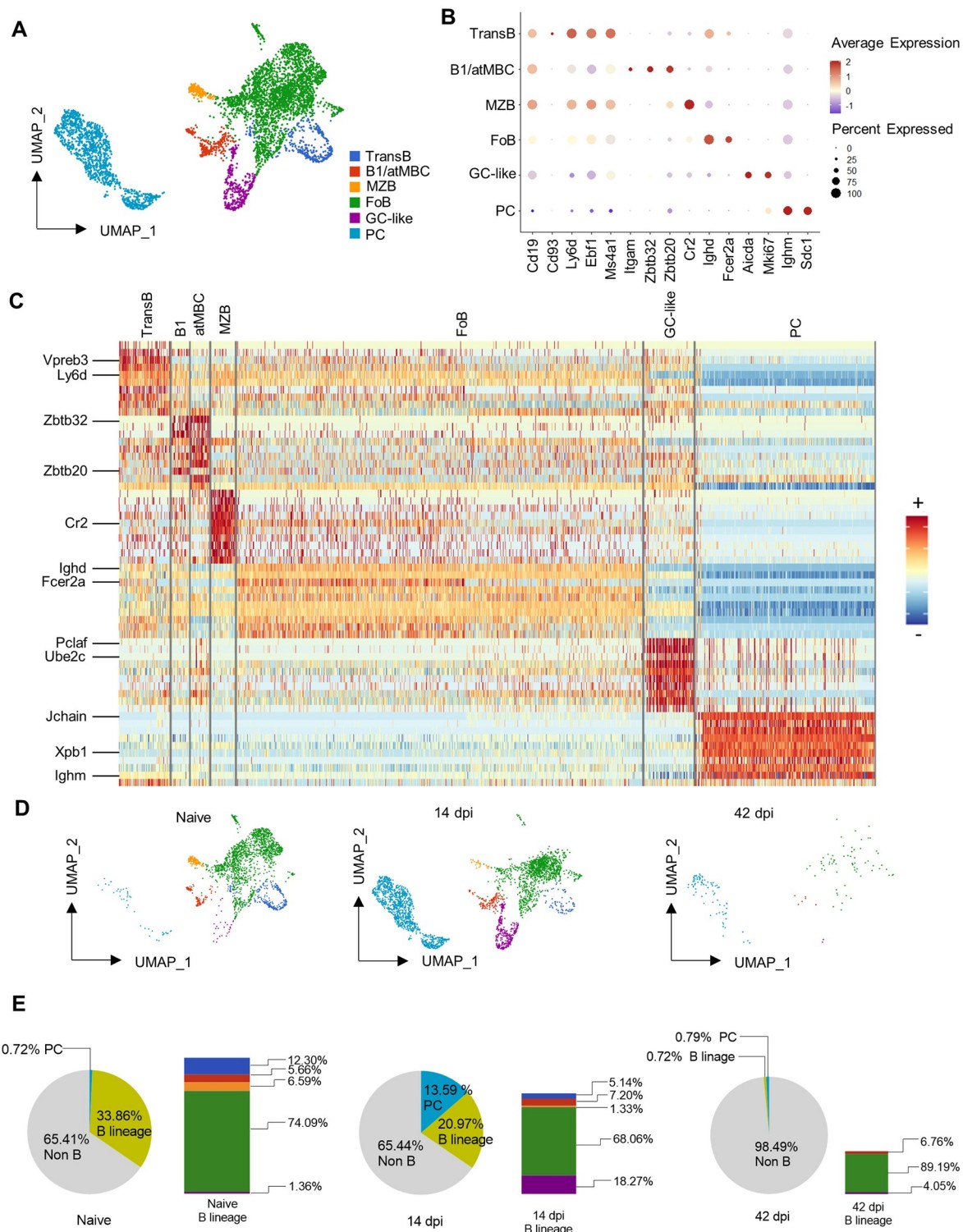

**Fig 2. Transcriptomic analysis reveals *T. evansi* induced terminal B cell differentiation. (A)** Uniform Manifold Approximation and Projection (UMAP) of B cells colored by annotated cell type. TransB: Transitional B cells, B1/atMBC: B1 cells/atypical Memory B cells, MZB: Marginal Zone B cells, FoB: Follicular B cells, GC-like: Germinal center-like B cells, PC: Plasma cells. **(B)** Gene expression of main gene markers used for annotating B cells. **(C)** Heatmap of combined 10 highest differential expressed genes (Wilcoxon Rank Sum test, Seurat) per each cell population. Only major genes are highlighted. See S4 Fig for full gene labeled list. **(D)** UMAP Split view of B cell populations across 3 time points. Color code corresponds to (A). **(E)** Pie charts: Percentage of each population in total splenocytes. Grey:

Non-B cells, Bright Yellow-Green: B lineage, Blue: Plasma cells. Bar graphs: Percentage of each B cell population in total splenic B cell lineage, colors correspond to (A).

by 42dpi. Bar graph representations (Fig 2E) show a rapid destruction of the Transitional (blue) and MZB cells (yellow), coinciding with the rapid increase in the B GC-like (purple) cells by 14dpi. However, at 42dpi a depletion of the entire B cell compartment is observed.

### 3. *T. evansi* infection induces a unique transcript signature in MZBs and FoBs

Using a B220/CD1d based gating strategy, flow cytometry analysis shows a major impact of *T. evansi* infections on both the MZB and FoB spleen populations (see Fig 1). This observation was confirmed by an alternative independent IgM/IgD gating approach (Fig 3A), and detailed transcriptomic analysis. Here, *CD1d1* and *Cr2* (CD21) were used to visualize MZB cells, while *Ighd* and *Fcer2a* (CD23) allowed identification of FoBs (Fig 3B). Comparative data between the non-infected sample and samples derived from 14 and 42dpi show the collapse of both populations, with MZBs being absent at the later time point (Fig 3C). A differential expression analysis of these samples demonstrates that for both B cell populations one of the top genes up-regulated in 14dpi cells is the *J chain*. In MZB, this is accompanied by the upregulation of the IFNγ inducible genes *Stat1*, *Ifi30* and *Ifi47*. In FoBs, 14dpi *J chain* expression is accompanied by the upregulation of *Ighg2c*, the IFNγ inducible gene *Socs3* and the gene coding for spleen light zone B cell activation marker CD83, indicating the initiation of GC activation. These observations are in line with infection-induced inflammation associated IgM and IgG2c production (Fig 1C). FoBs showed down-regulation of *Fcer2a* (CD23) at 14dpi, confirming the flow cytometry analysis (Figs 1E and 3A). Further Gene Ontology analyses of top 100 differentially expressed genes in MZB/FoB populations at 14dpi show that both populations express elevated levels of transcripts coding for proteins involved in B cell activation, antigen processing and presentations, protein folding and expression, as well as cellular response to interferons (Fig 3D and 3E right panels). The full list of differential expressed genes resulting from DEG analysis is presented in S1 File.

### 4. Loss of bone marrow B cell replenishment and a transient appearance of atypical memory B cells hallmarks early *T. evansi* infection

Transitional B cells are a minor population that was captured by scRNA sequencing. A combination of previously reported multiple signature gene markers [31,32] was used to identify and visualize this population (Fig 4A and 4B). Transitional B cells were annotated based on the high expression of the early-stage B cell markers CD93 and Ly6D, the pre-B-cell receptor surrogate light chain encoding gene *Vpreb3*, and *Ighm* encoding the IgM heavy chain, all in the absence of IgD (*Ighd*). This population undergoes a very rapid size collapse during infection, showing a 4-fold reduction by 14dpi. These cells were virtually absent at 42dpi. To understand the rapid loss of this population, bone marrow samples of non-infected and infected mice were analyzed. Six weeks into infection, mice showed a nearly complete ablation of the CD93+ early linage B cells in the bone marrow, while these cells should under homeostatic conditions be present in order to replenish the periphery (Fig 4D). ScRNAseq analysis also showed the rapid loss of spleen B1 B cells. These cells have so far never been described in the context of trypanosomosis, most likely because the analytic resolution of flow cytometry is insufficient to visualize this minor cell population during infection. However, here we were able to identify these cells in the naïve spleen sample using the specific *S100a6* marker in combination with

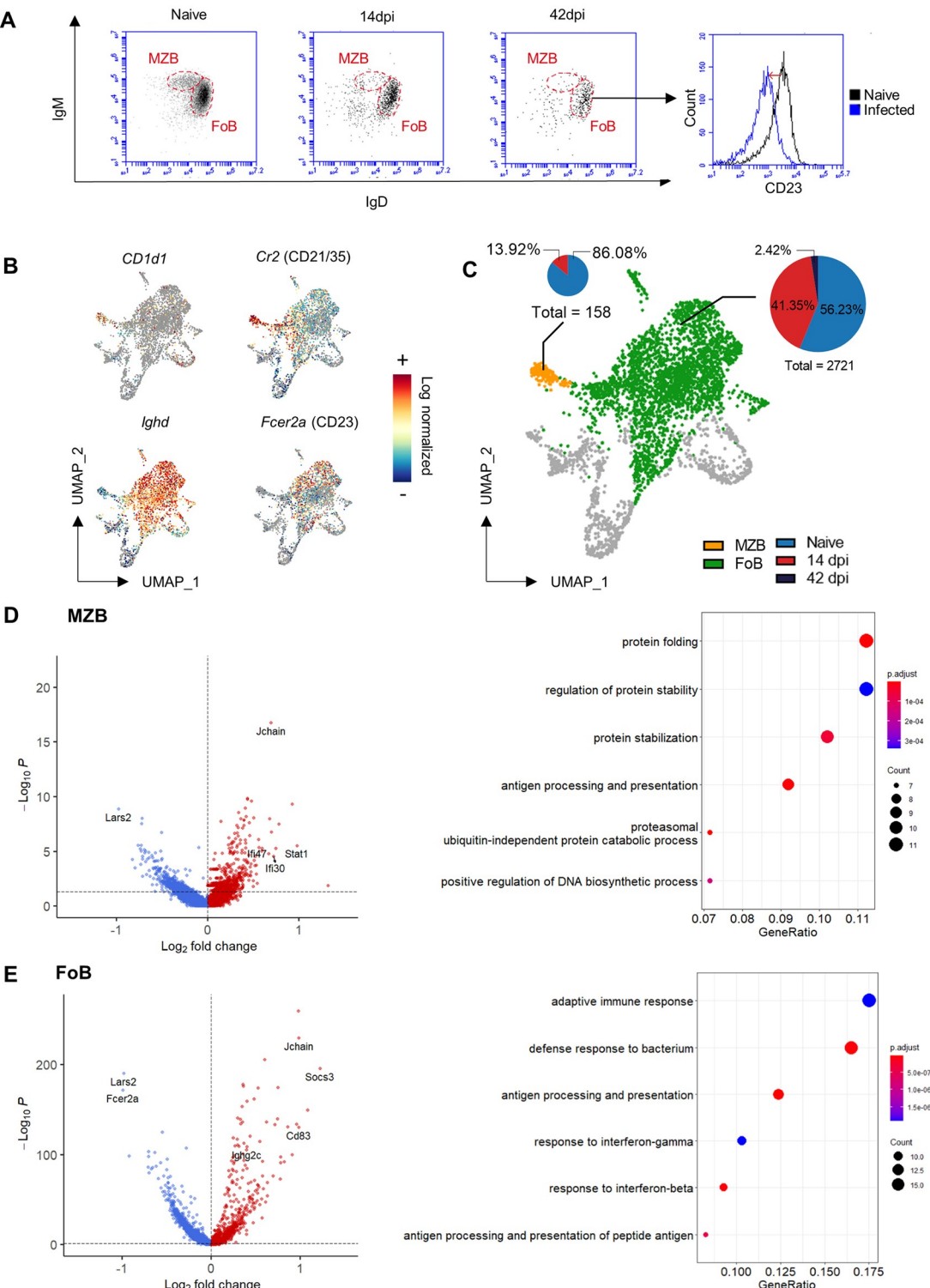

**Fig 3. Differential gene expression analysis of MZBs and FoB during _T. evansi_ infection. (A)** IgM/IgD flow cytometry profile of MZB and FoB during infection (left panels) and FoB-CD23 expression histogram plot during infection (right panel). One representative data set of 9 individual measurements is shown. **(B)** Log normalized expression of _Cd1d1_, _Cr2_, _Ighd_, and _Fcer2a_ in spleen B cell populations in UMAP projection. **(C)** MZB and FoB populations UMAP visualization, with proportional pie-chart representation of population at 3 different time points. Pie-chart diameters are normalized for each population size. **(D-E)** Left panels: Differential gene expression analysis of MZB/FoB at 14dpi versus Naïve. Dash lines display cut-off of P value (0.05). Right panels: Gene ontology analysis of top 100 differential expressed genes of MZB/FoB at 14dpi.

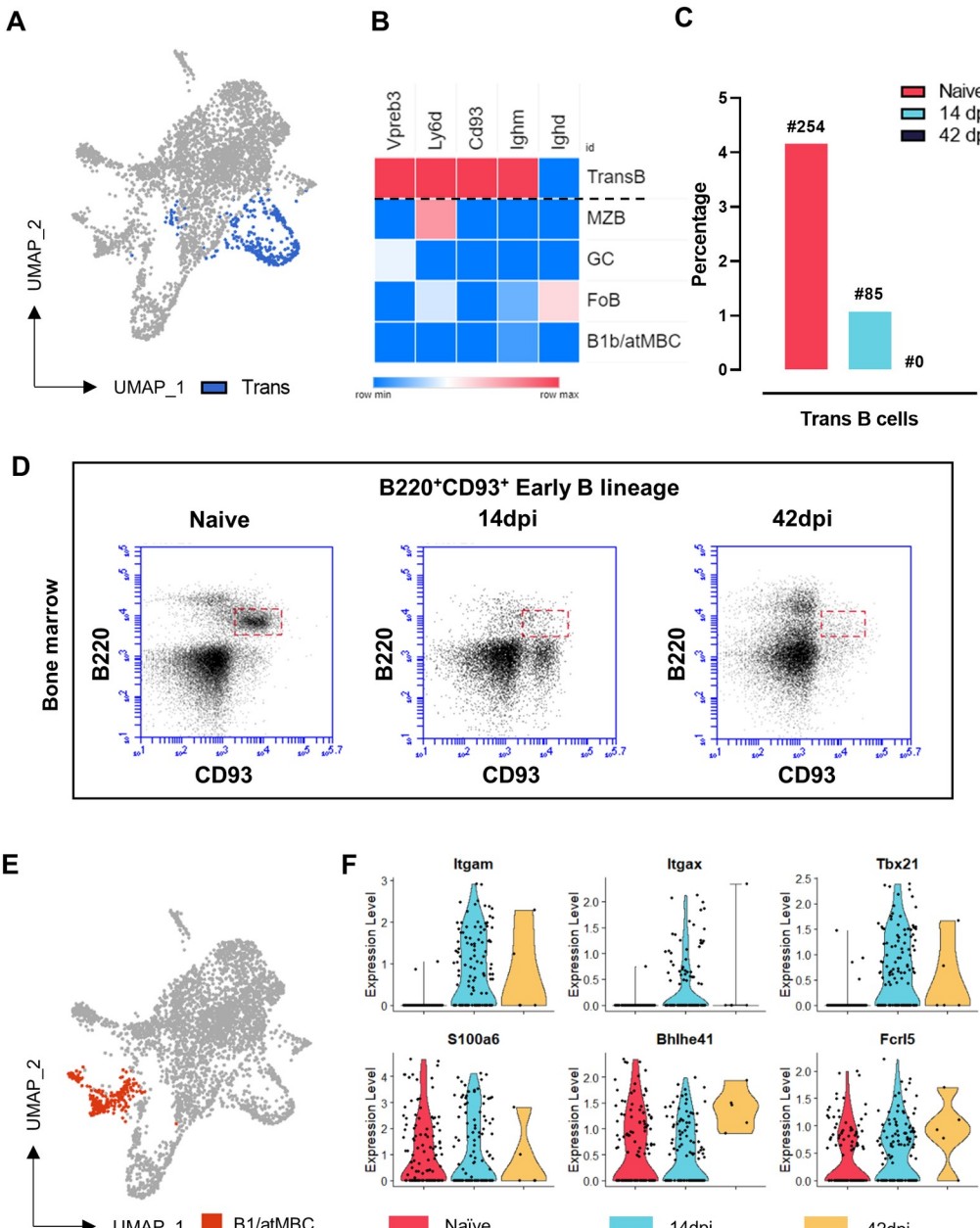

**Fig 4. Lack of a spleen replenishment by early B lineage cells, reduction of B1 cells and a transient appearance of atypical memory B cells. (A)** Transitional B cells population location in UMAP projection. **(B)** Z-score expression of Transitional B cells gene makers across B cell populations. **(C)** Percentage of splenic Transitional B cells at 3 different time points, indicating absolute cell number at each time point. **(D)** Early B lineage cells in bone marrow by flow cytometry analysis at 3 different time points during infection. Results are a representative data of a triplicate repeat experiment with 3 mice. **(E)** B1/atMBC population location in UMAP projection. **(F)** Expression of signature genes by atMBCs (upper panel) and B1 cells (lower panel) in Naïve, and infected mice 14dpi and 42dpi. Black dot represent a single cell.

*Bhlhe41* and *Fcrl5*. Interestingly, while this population was reduced in size at 14dpi (and virtually absent at 42dpi), it was replaced in the UMAP projection at 14dpi by a populations carrying the signature of short-lived atMBCs, being CD11b⁺ (*Itgam*), CD11c⁺ (*Itgax*) and T-bet⁺ (*Tbx21*) (Fig 4E and 4F).

## 5. GC-like B cell induction occurs in the absence of B follicles

Despite previous reports demonstrating trypanosomosis-associated destruction of splenic B cell follicles and overall tissue architecture [21], scRNA-seq analysis shows that *T. evansi* induces GC-like B cells marked by *Aicda* expression, crucial for the CSR and immunoglobulin affinity maturation (Fig 5A, left). High expression of *Pcna*, involved in SHM, as well *Mki67*, a key marker for cell proliferation, confirm this annotation (Fig 5A, middle/right). Combined transcriptomic data analysis demonstrates that GC-like cells expand rapidly during infection with majority of these B cells derived from the 14dpi sample (Fig 5B). The presence of infection-induced GC-like B cells at 14dpi was confirmed by flow cytometry analysis (Fig 5C left), and immunohistochemistry (Fig 5C, right). These data show for the first time that experimental trypanosomosis in mice results in a peak of the GC-like B cells two weeks into infection, despite the destruction of B follicles as a result of infection-derived splenic structure collapse (Fig 5D). Subsequently, PCs differentiation was assessed. Here, *Sdc1* (CD138), *Xbp1* (a master regulator of the unfolded protein stress response pathway crucial for PC development) and *Pax5* (a B cell lineage marker downregulated in GCs B and absent from PCs) were used as gene markers for annotation (Fig 5E, left). Further sub-classification was done based on expression of *Ighm* (IgM), *Ighg2c* (IgG2c) and *Igha* (IgA) (Fig 5E, right), coding for immunoglobulin heavy chains. The numerical scRNA-seq data (Fig 5F) show the highest number of PCs expressing *Ighm* and *Ighg2c* transcripts observed on day 14 of infection, followed by the population collapse 6 weeks in to the infection, corroborating observations by flow cytometry (Fig 1E).

## 6. Blocking AID activity results in a very significant improvement of *T. evansi* elimination

The combined data above indicates an early induction of antibody secreting cells and an IgG2c isotype switch after the first peak *T. evansi* parasitemia. Unfortunately, this response does not benefit the host as parasites rapidly proliferate despite the Ab isotype switch. In order to assess how parasitemia would be controlled in the absences of this very early stage isotype switch event, a *T. evansi* infection was performed in AID$^{-/-}$ mice, unable to undergo Ig class switching or affinity maturation. Surprisingly, AID$^{-/-}$ mice exhibit a significantly improved early-stage trypanosomosis control, characterized by the virtual absence of circulating parasites during the first three weeks of infection, and a first-peak parasitemia level that is 30-fold lower than the peak observed in fully immune competent mice. In fact, half of the infected AID$^{-/-}$ mice showed parasite numbers persistently below the detection limit of a hemocytometer, being only occasionally detectable in undiluted thick-smear preparations (Fig 6A). Interestingly however, once parasitemia started to increase, the infection quickly reached lethal heights, resulting in the observation that the infected AID$^{-/-}$ mice did not have any significant difference in survival as compared to fully immune competent mice. The absence of IgGs in these mice however did not have any effect on survival as compared to fully immune competent mice (Fig 6A compared to Fig 1). Coinciding with the improved parasitemia control, AID$^{-/-}$ mice had natural IgM Ab titers against VSG that were measurable even in 1/1600 diluted plasma prior to infection. These titers remained stable during the 1$^{st}$ week of infection but significantly increased by 14dpi, reaching a 1/12.800 endpoint titer. In contrast, fully immune competent wild type mice (WT) only showed VSG binding IgM antibodies at 14dpi, with a 1/3200 endpoint titer, 4x lower than the infected AID$^{-/-}$ mice (Fig 6B upper panels). As expected, only the WT mice showed anti-VSG IgG2c antibodies (1/3200 end point titer), induced after the first peak parasitemia (Fig 6B lower panels).

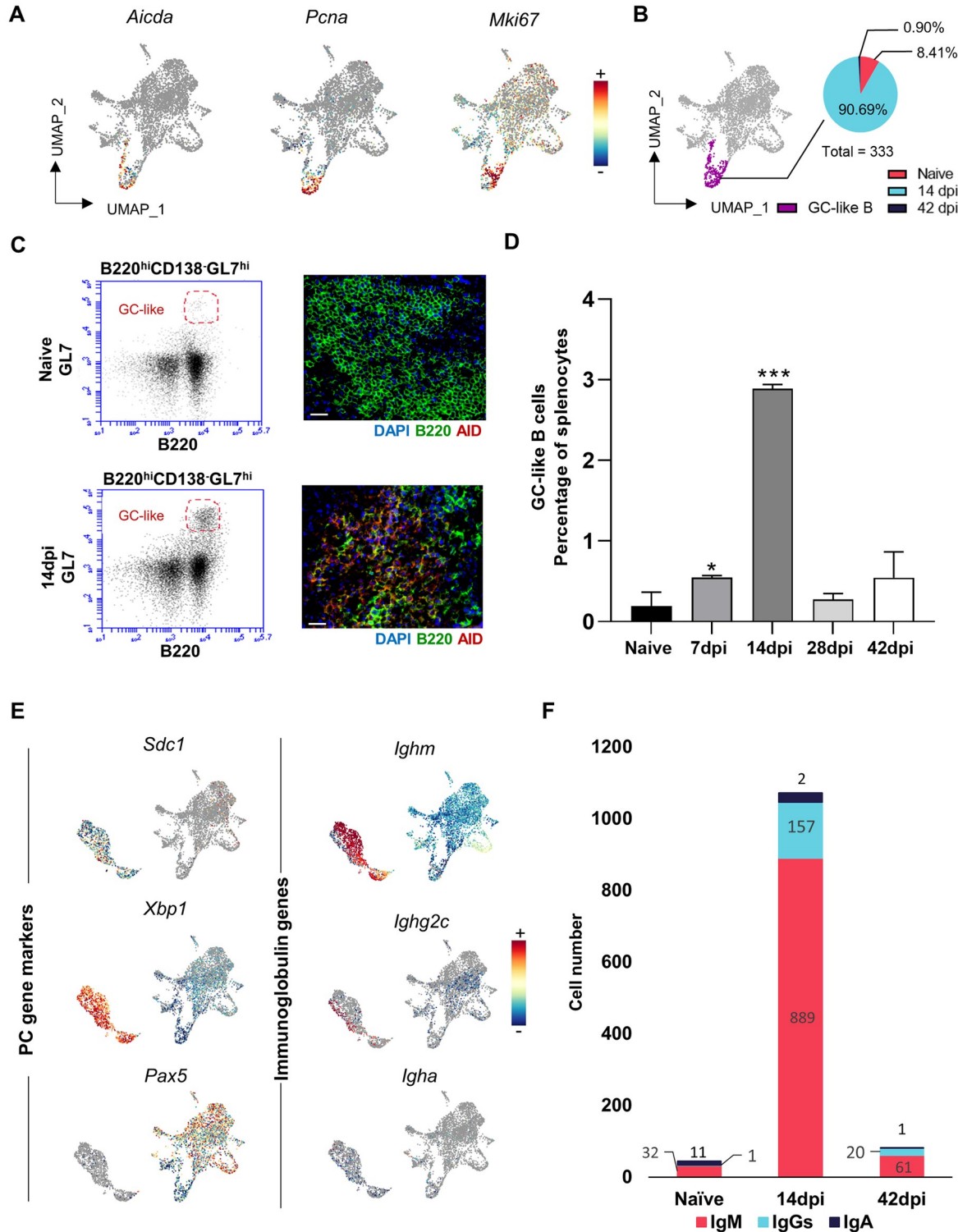

**Fig 5. Rapid formation of GC-like B cells and PCs results in decline of cell numbers later during infection. (A)** UMAP projection of B cells colored based on log normalized expression of *Aicda*, *Pcna* and *Mki67*. **(B)** GC-like B cells UMAP visualization, with proportional pie-chart representation of population at 3 different time points. **(C)** GL7/B220 flow cytometry analysis (left) and AID/B220 immunohistochemistry of infection-induced GC-like B cells at 14dpi. Scale bar indicate 20 μm. **(D)** Percentage of GC-like B cells in splenocytes during *T. evansi* infection, analyzed by flow cytometry analysis. Data is shown as mean + SD of three independent samples, from one of three representative experiments, with significant differences versus naïve controls. * p≤0.05; ** p≤0.01; *** p≤ 0.001 by

Student's t-test. **(E)** B cells and PCs colored based on log normalized expression of *Sdc1*, *Xbp1*, *Pax5* (left) and *Ighm*, *Ighg2c*, *Igha* (right). **(F)** Distribution of splenic IgM⁺ PCs, IgGs⁺ PCs, and IgA⁺ PCs at 3 different time points.

To corroborate a direct anti-trypanosome activity of the AID⁻ᐟ⁻ IgM antibodies, *T. evansi* parasites were cultured *ex vivo* for 4 hours in the presence of 10% WT or AID⁻ᐟ⁻ mice plasma at 3 different time points: naïve, 7dpi and 14dpi. The results show for naïve and 7dpi that plasma derived from AID⁻ᐟ⁻ mice exhibit significant superior detrimental effect on parasite survival compared to WT mice plasma (Fig 6C). In fact, WT mice plasma only show significant lytic activity at day 14 post infection (Fig 6C). Additionally, an *in vivo* transfer setup shows that 7dpi AID⁻ᐟ⁻ plasma but not WT plasma has protective anti-trypanosome activity in fully immune competent mice (Fig 6D).

Conceding with the superior parasitemia control, *T. evansi* infected AID⁻ᐟ⁻ mice are characterized by the preservation of their spleen B cell compartment, showing the full retention of FoBs and an improved maintenance of the MZB compartment (Fig 7A and 7B and S5A Fig), as well as preservation of the bone marrow pool of immature B cells (S5B Fig, lower panel). This was accompanied by a reduced differentiation into PCs as compared to the WT mice (Fig 7A and 7B). Finally spleen architecture was also much better maintained with the presence of B follicles in infected 14dpi AID⁻ᐟ⁻ mice (Fig 7C). As such, results indicate that high levels of IgMs present early on during infection, are correlated to the protection observed in these mice, while the AID-driven GC-like B cell differentiation into IgG2c producing PCs observed in immune competent mice exerts no beneficial effects on outcome of parasitemia control.

## Discussion

*Trypanosoma evansi* is an extracellular infectious agent with a near-worldwide geographic distribution and potential human infectivity [3,15]. In livestock, it causes failure of commercial vaccines against non-related bacterial of viral diseases [26]. To be successful, trypanosomes have to diminishing the risk for antibody mediated killing. For this, they use antigenic variation of the surface coat [9,33], induction of B cell and T cell dysfunctions [17,21,34,35], inhibition of complement activation [12] rapid clearance of surface bound antibodies [11] and even quorum sensing [36]. While these mechanisms have mainly been studied using laboratory adapted *T. b. brucei* parasites, limited immunological evidence is available in relation to *T. evansi*. Hence, our approach focused on the parasite-host interaction of the latter, and components of the immune system that limit parasite growth, with emphasis on crucial B cell populations and antibody-mediated infection control. This study provides for the first time a trypanosomosis-derived comprehensive set of transcriptomic scRNAseq data, in combination with flow cytometry, immunohistochemistry, histology and serology results. Together, they suggest that the rapid induction of IgG2c during the early onset of infection serves as a decoy mechanism to evade antibody mediated killing by innate B cells. The latter produce poly-reactive trypanocidal IgMs that can control parasitemia independently of CSR and SHM.

Natural antibody mediated immunity against bloodborne pathogens is mainly provided by B1 B cells and MZBs [37,38]. The latter are B220⁺CD21^high IgM⁺ mature B cells, capable of providing rapid T cell-independent IgM responses and delivering long-lived IgM PCs that can contribute to long-term immunity [39,40]. While it has been shown in various trypanosome infections that MZBs rapidly reduce in numbers [19–21], the impact of *T. evansi* infection on these cells was never studied. Hence, here we show that *T. evansi* induces the rapid reduction of MZB cell numbers through a process that carries the signature of IFNγ activation and cell differentiation into both short-lived atMBCs and IgM producing PCs. This observation falls in line with our earlier findings, i.e. that IL-12 dependent interferon-γ induction is crucial for *T.*

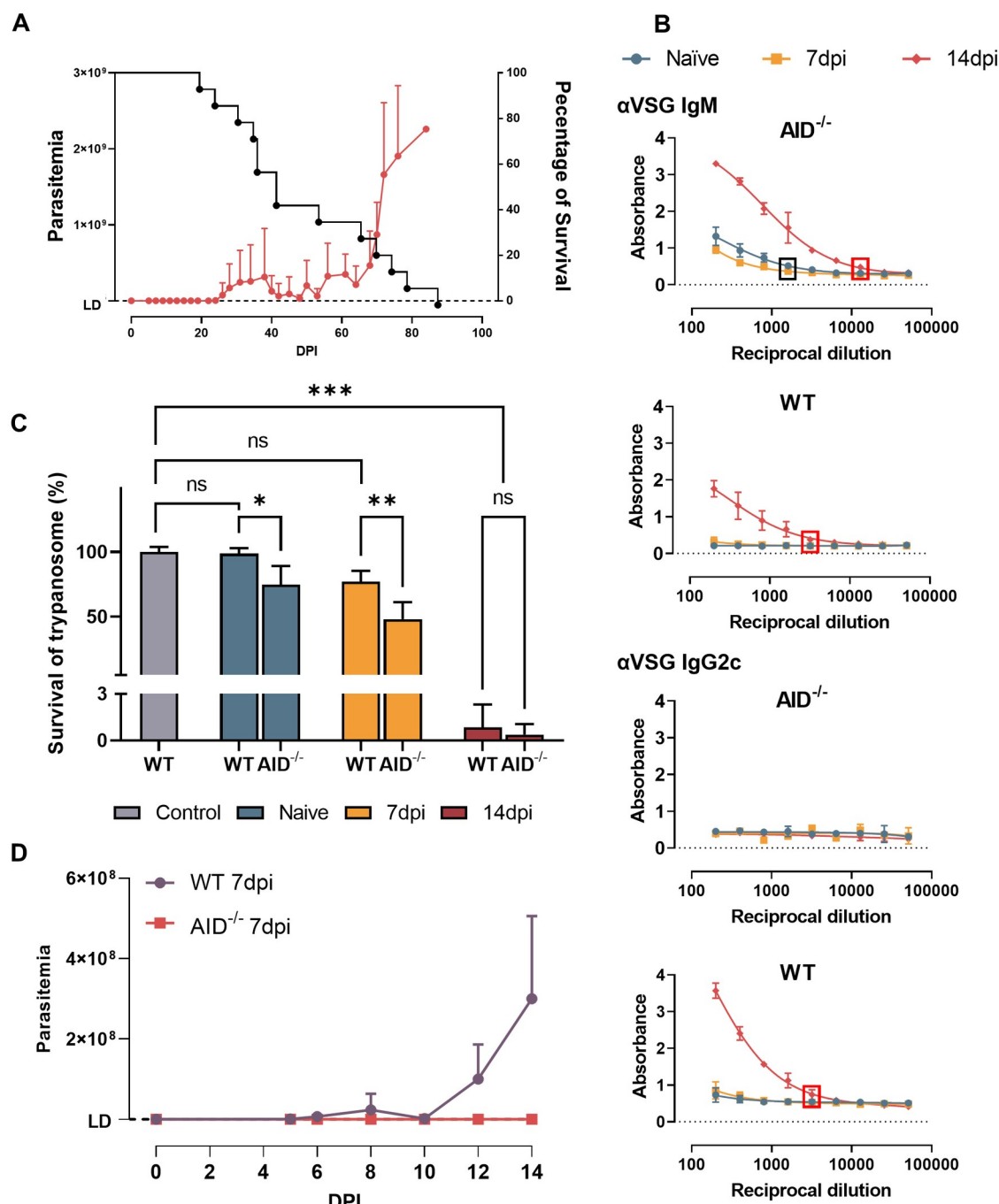

**Fig 6. AID<sup>-/-</sup> infected mice exhibit superior trypanosomosis control.** (A) Parasitemia and survival of *T. evansi* infected AID<sup>-/-</sup> mice (means + SD of 15 mice, combining 3 independent experiments). The Left Y axis indicates the number of parasites while the right Y axis indicating percentage of survival. LD: detection limit ($2 \times 10^5$ parasites/ml). **(B)** Anti-VSG IgM and IgG2c isotype serum antibody titers in infected AID<sup>-/-</sup> and WT mice (right). The box highlights the end point titer (black = naïve and 7dpi, red = 14dpi). **(C)** *In vitro* trypanocidal activity of AID<sup>-/-</sup> and WT mice plasma at different time points. Control bar indicate survival of trypanosome in HMI9 media without plasma incubation. * p≤0.05; ** p≤0.01; *** p≤ 0.001; ns: non-significant by One-way ANOVA followed by multiple comparisons. **(D)** Anti-trypanosome activity of AID<sup>-/-</sup> and WT 7dpi mice plasma after transfer to WT mice 3 days after infection. LD: detection limit. **(B-C)** Data is represented as means + SD of three individual samples, in one of three representative experiments.

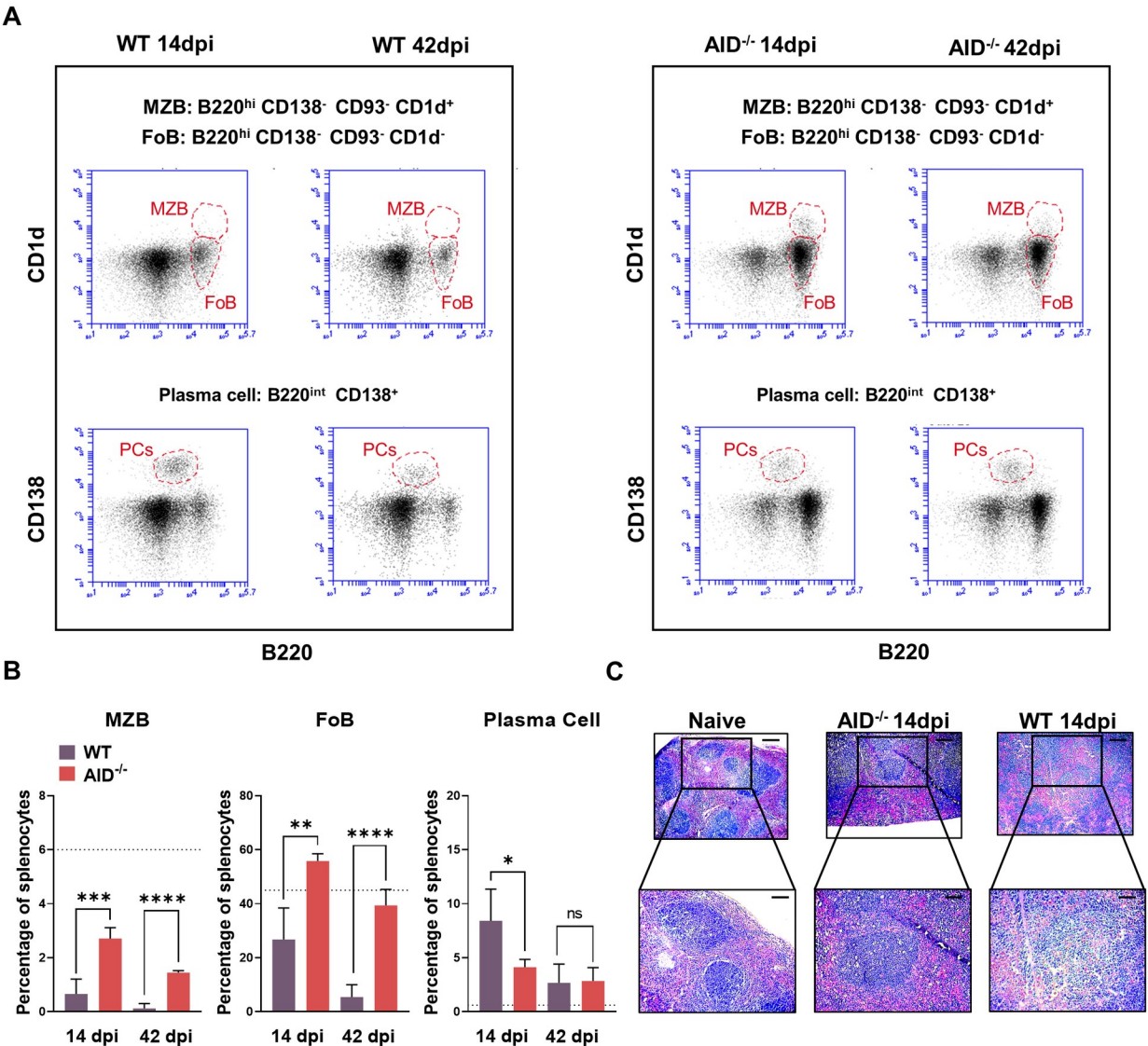

**Fig 7. AID$^{-/-}$ infected mice are characterized by superior maintenance of the splenic B cell compartment. (A)** MZB, FoB and PCs of 14dpi and 42dpi *T. evansi* infected WT (left) and AID$^{-/-}$ (right) mice. One representative mouse (of a total of 9 individual measurements) is shown for each group. **(B)** Percentage of MZB, FoB and PCs in splenocytes during infection. Data is shown as means + SD of three individual samples, in one of three representative experiments, with significant between AID$^{-/-}$ infected mice and WT infected mice calculated by Student's t-test. $^{*}$ p≤0.05; $^{**}$ p≤0.01; $^{***}$ p≤ 0.001. Dash line displays percentage of the population in spleen of naïve WT/AID$^{-/-}$ mice as both of them show non-significant difference in cell portion at naïve stage (see S5A Fig). **(C)** Spleen section H&E staining (scale bar indicate 250 μm for top panels and 100 μm for bottom panels).

*evansi* control [41]. The cell activation process is hallmarked by expression of key IFNγ inducible B cell genes *Stat1* [42], *Ifi30* and *Ifi47*, as well as the *J chain*. The *Ifi30* gene encodes a lysosomal thiol reductase linking innate immunity with MHC class-II restricted antigen processing [43], while *Ifi47* is involved in early B cell activation [44]. The J chain on the other hand is not only required for correct IgM assembly, but also closely interacts with the regulation of Blimp and Pax5, both involved in PC differentiation [45]. Hence, the new data suggest that 'disappearance' of MZBs in trypanosomosis is not mainly due to cell death as previously

proposed in a *T. brucei* model, but can be explained by proliferative differentiation of MZBs into IgM producing PCs [40].

Interestingly, as MZB numbers rapidly declined following the first peak of infection, so do splenic innate B1 cell number. This population was annotated using *S100A6* as a gene marker for differentiation of these cells from the major B2 spleen cell B cell population. This Annexin A2 binding protein coding gene has previously been shown to be upregulated during the B1 activation process [46]. In addition, high expression of the *Zbtb20* transcription factor allowed distinguishing these cells from the very similar atMBCs [47]. The *Bhlhe41* and *Fcrl5* genes on the other hand have been either linked to the generation of B1 B cells [48] or the counter regulation of BCR signaling in innate B cells [49]. So far, there have been no reports on spleen B1 B cell function in experimental mouse trypanosomosis, possibly because of the difficulty to observe these cells in flow cytometry. In contrast, in *T. evansi* infected sheep and *T. congolense* infected cattle the presence of CD5$^+$ B1 cells has been reported, indicting the host-specific involvement of these innate cells in trypanosome immunity [50,51]. Interestingly, mouse B1 B cells are activated by CpG DNA, through TLR9 engagement, driving T cell independent immunity [52–54]. TLR9 engagement is involved in trypanosomosis associated IFNγ induction as well as parasitemia control of *T. b. rhodesiense* [55]. As TLR/MyD88 signaling gives rise to VSG-specific antibody induction [56], and TLR signaling appears crucial for maintenance of B cell memory [57,58], CpG DNA has even been proposed as an adjuvant in anti-trypanosome vaccine approaches [59–62]. Finally, it is worth nothing that IgM$^+$ B1 B cells have also been shown to play an important role in malaria-associated antibody production [63] as well as *M. tuberculosis* associated IgM secretion [64].

In mammals, FoBs represent the major B cell population that is 'ready' to initiate a long-lasting adaptive antibody response to invading pathogens. Hence, from an evolutionary point of view, it is critical for extracellular trypanosomes to be able to deal with this threat. For FoB characterization, CD23 is most often used as the discriminatory marker in flow cytometry. However, rapid downregulation of gene and surface CD23 expression is a signature feature of experimental trypanosomosis, and coincide with the switch of FoBs towards IgG2c production. With CD23 being a negative regulator of B cell receptor signaling, it acts as an inhibitor of BCR clustering and phosphorylation of Btk. Hence, CD23 KO mice exhibit increased IgG responses to protein antigens [65–67], in line with our observations. For FoBs, immune activation during the first weeks of infection is marked by the upregulation of the IFNγ inducible gene *Socs3* and the spleen light zone B cell activation marker CD83 [68]. The Gene Ontology analysis indicates both IFNβ and IFNγ signaling, as well as the activation of antigen processing and presentation. While these findings do not exclude that part of the FoBs is eliminated from the spleen in an NK dependent manner, as has been shown to occur during *T. brucei* infections [18], our scRNA-seq data does not indicate that this is the major pathway explaining the 'collapse' of the population. Our results rather points to the rapid differentiation into antibody producing cells. Interestingly, while there is ample literature on the crucial role of IFNγ in control and pathology of trypanosomosis [41,69–74], the role of an IFNβ mediated response is intriguing as IFN type-I immune protection is conventionally linked to viral immunity [75]. So far, there are no data available on the role of IFNβ in the control of trypanosomosis, but increased IFNβ transcription is found in the brains of *T. brucei* infected mice, and IFNAR signaling is involved in the infiltration of trypanosomes into the brain parenchyma [76,77]. IFNAR$^{-/-}$ mice exhibit an impaired first peak parasitemia clearance during *T. b. rhodesiense* infections [76]. Hence, it appears that the role of Type-I interferons in trypanosomosis is complex and needs to be further investigated, in particular as it can impair T-dependent responses through the destruction of B cell follicles, resulting in reduction of affinity matured antigen-specific antibody responses [78].

As previously described, most anti-trypanosome Ab response appear to result from T cell independent process. This has been shown in experimental *T. brucei* models for the induction of rapid IgG2a/c responses [79], and could also explain the rapid occurrence of CD11b[+] CD11c[+] T-bet[+] short-lived atMBCs. The latter were only observed at 14dpi and exhibited a phenotype that is very similar to that of innate B1 B cells. So far, these cells have been associated with protective IgG2a/c anti-viral responses [80], and autoimmune responses [81]. Hence, during *T. evansi* infections atMBCs could not only contribute to the rapid IgG2c anti-trypanosome response, but could also give rise to the autoimmune complications, a signature pathology of trypanosomosis [82]. In general, high-affinity antibody maturation and class switching occur mostly in GCs, in a T cell dependent manner. However, T cell independent GC formation can occur [83] and alternatively, extra-follicular IgG class switching can deliver high affinity IgG2c antibodies [84]. As experimental trypanosomosis results in the loss of spleen follicles and the absence of GC formation [21], no data has been reported on the role of GC B cells in this setting. Moreover, Lymphotoxin-α deficient mice that lack any GC formation, have been shown to control *T. brucei* trypanosomosis in a similar way as fully immune competent WT mice, with high IgG2c anti-VSG tires early-on in infection [79,85]. In our study, the scRNA-seq data analysis demonstrated the very clear induction of a distinct cell population carrying all hallmarks of GC B cells, including upregulation of *Aicda*, allowing for rapid SMH and CRS to take place [86,87]. These cells showed upregulated expression of genes associated with proliferation such as *Pclaf* [88] *Pcna* [89] and *Mki67* [90,91], as well as the cell cycle progression marker *Ube2c* [92]. Subsequent GL7-based flow cytometry analysis confirmed the rapid accumulation of GC-like B cells in the spleen, while in contrast, histological staining showed the absence of actual B follicles in *T. evansi* infected mice. This finding relates to the trypanosome CpG-DNA immune activation outlined above, as both class switch recombination and somatic hyper-mutation require the consorted action of both BCR and TLR signaling and can occur in the presence of the IFNβ mediated destruction of GCs [93,94].

*T. evansi* induced B cell maturation and differentiation ultimately leads to PC formation by 14dpi, characterized by highly expressed *Xbp1* in the absence of *Pax5* [95]. The vast majority of these cells are IgM[+], while a smaller fraction is IgG2c[+]. These findings explain early-stage induction of IgM/IgG2c plasma titers during infection. Yet, to understand the final collapse of the host B cell compartment and loss of subsequent infection control, attention should be drawn to (i) the short lived nature of these infection-induced PCs, and (ii) the rapid loss of the early lineage CD93[+] B cells from the bone marrow of infected mice, as well as the loss of Transitional splenic B cells [31,32]. These phenomena have already been described in detail in *T. brucei* infections, and result mainly from infection-induced inflammatory apoptosis [17]. We conclude here that the functional consequence of this depletion is that once a terminal GC/PC differentiation takes place in a response to the first peak of a *T. evansi* infection, the periphery is not being replenished with naïve B cells, impairing the host to mount efficient subsequent antibody responses against newly arising trypanosome variants.

Combined, the data discussed above indicate that trypanosomes drive an early-stage IgMs/IgG2c antibodies switch and terminal PC differentiation, followed by the gradual depletion of the host B cell compartment due to the lack of B cell replenishment. Ultimately, this causes a loss of parasitemia control and death of the host. Hence, successful infection of the host by trypanosomes requires the capacity of the parasite to evade population eradication during the first week of infection, driving an inflammatory pathology that is mainly accompanied by T cell independent and presumably low-affinity IgG2c antibody induction. This observation triggered us to investigate how *T. evansi* infections would be controlled in an environment with increased levels of natural high-avidity IgM antibodies, and an inability to undergo class switch recombination. Hence, *T. evansi* infection was performed in AID[-/-] mice, marked by

increased level of natural IgM antibody titers [86,96,97]. As reported here, these mice showed a surprisingly and previously undocumented superior level of trypanosomosis control. This control coincided with maintenance of the IgM$^+$ MZB compartment, the early lineage bone marrow B cell pool, splenic architecture, and sustained elevated serum IgM titers in the absence of any IgGs. Interestingly, plasma of naive AID$^{-/-}$ mice containing high IgM levels was shown to bind to VSG in ELISA and exert trypanocidal activity *in vitro*. In addition, transfer of plasma from 7dpi infected AID$^{-/-}$ mice into WT infected mice abrogated *T. evansi* infection onset, in contrast to 7dpi plasma of WT mice. These data confirm our previous results showing that purified infection-induced IgM fraction, but not IgGs, can provide sterile protection against a homologous challenge in naïve recipient mice in an in *vivo* antibody transfer assay [98].

Finally, the observations presented here require to speculate on the functional mechanisms of the superior IgM mediated trypanosomosis control. We have previously showed that IgM$^{-/-}$ mice are characterized by impaired trypanosomosis control [98,99], and IgMs, which are known to be superior when it comes to complement activation as compared to IgGs, have been shown to be crucial for trypanosome phagocytosis by CR3$^+$ macrophages using an *in vitro* setting [100]. Interestingly, recent *in vivo* data has shown that it is the CRIg receptor expressed on Kupffer cells, and not the CR3 receptor on macrophages that plays a crucial role in the intravascular clearance of *T. congolense* parasites [101]. However, the true mechanism of antibody-mediated trypanosome clearance is still a matter of debate, especially as single chain Fc-devoid nanobodies are very efficient in killing trypanosomes [102,103]. This only required high-affinity surface binding, subsequently slowing down VSG surface recycling, leading to motility arrest and rapid cell death. As such, while IgGs might have a higher binding affinity as compared to IgMs, the latter have an improved binding avidity due to their decavalent binding pattern and unique epitope recognition [104]. While VSG-antibody interaction modeling shows that when one IgG 'VH/VL arm' binds to VSG one homodimer, the other 'arm' can bind to any of 18 other VSG homodimers on the parasite surface [105], this number is increased to 9 IgM 'arms' being able to bind any of 61 other VSG homodimers, tremendously increasing avidity [106]. *In vitro*, this results in an accelerated antibody clearance of IgM compared to IgG in a pulse-chace setting [11], but in an *in vivo* setting, the final biological availability of surface bounds Abs will strongly depend on the serum concentrations of molecules as well as their production rate by PCs. This would give AID$^{-/-}$ mice an advantage with improved antibody-mediated phagocytosis and parasite killing due to the high production rate of natural IgMs. In this context, we hypothesize that IgG2c in WT mice could eventually compete with IgM for surface binding, hence reducing the efficacy of IgM-mediated phagocytosis and/or complement activation. This would not be a unique situation for trypanosomes as also in other protozoan infections such as malaria and leishmaniosis, IgMs play a crucial role [107,108]. Long-lived IgM responses have also been shown to deliver good anti-viral responses [109–111], a finding that now is receiving renewed attention due to the COVID-19 crisis [112].

Taken the crucial role of IgMs in trypanosomosis, our new results expose a dilemma for disease control strategies. How can one develop a long-lasting IgM vaccine with memory recall activity, without reducing the levels of protective IgM titers while immune maturation progresses through CSR and SHM? So far, anti-trypanosome vaccine research has always focused on finding good immunogens, capable of inducing class-switched antibodies, and in some cases memory B cells [14]. The new AID$^{-/-}$ results could explain why this approach has never yielded a field-applicable solution with proper memory recall activity. If indeed trypanosomes are well adapted to deal with IgGs, and are in large only controllable by natural IgMs and MZB, they might have acquired all evolutionary tools necessary to evade most vaccine

strategies available today. There is a simple explanation as to why trypanosomes would have evolved this way: as antigenic variation avoids high-affinity interactions with antibodies raised against successive waves of parasitemia, it would not provide invisibility to a maturing SHM antibody response, supported by the presence of more conserved T helper cell epitopes. Hence, if affinity matured IgGs were to be detrimental to the trypanosome, over time the immune system would be able to mount a broad range IgG response to the vast majority of VSGs, leading to the eventual elimination of the infection. In contrast, trypanosomes are ill adapted to fight an AID$^{-/-}$ IgM-locked immune environment. Here, parasites are incapable of driving rapid T cell independent polyclonal TLR-mediated IgG2c class switching and while IgM$^{+}$ MZB cell numbers do gradually decline during infection in AID$^{-/-}$ mice, this process progresses much slower compared to WT mice. Hence, we conclude that by rapidly inducing IgGs that may compete for antigen binding with natural IgMs in immune competent hosts, *T. evansi* has adopted a niche solution that allows them to thrive in 'plain sight' of the adaptive antibody response.

## Materials and methods

### Ethics statement

Experimental animal procedures were approved by the Ghent University Global Campus Institutional Animal Care (GUGC IACUC), allowing for parasitemia follow-up, organ harvesting and survival time monitoring, taking into account human endpoint determinations (IACUC 2020-008/016/020/021). Use of AID$^{-/-}$ mice was granted for comparative data analysis during the first 6 weeks of infection (IACUC 2021–006) and parasitemia/survival time monitoring (IACUC 2020–021).

### Experimental model and subject details

**Mice.**    Female 7–9 week old C57BL6/N were purchased from Koatech, Korea and AID$^{-/-}$ mice (B6.Cg-Aicda<tm1Hon> (N10)/HonRbrc) were provided by Riken Research institute, Japan [86]. All mice were housed in IVCs (individually ventilated cages) and were provided appropriate cage enrichment.

**Parasite and infection.**    *T. evansi* Merzouga 93 (ITMAS 150399C), originated from Morocco, was obtained from the Institute for Tropical Medicine, Antwerp, Belgium. Infection was initiated by intraperitoneal (i.p.) injection of 200 *Trypanosoma evansi* Merzouga 93 parasites. Parasitemia was assessed in blood collected from the tail vein, at different time points during infection. Blood was diluted (1/200) in Dulbecco's Phosphate Buffered Saline (DPBS; Invitrogen, CA, USA) and the number of trypanosomes present in the blood was counted using a hemocytometer and a light microscope.

### Method details

**Cell isolation and flow cytometry analysis.**    Spleen from naïve and infected mice were isolated at different time points. Single-cell suspensions were prepared by homogenizing spleens in 6 mL of DMEM (Capricorn Scientific, Ebsdorfergrund, Germany) supplemented with 10% FBS (Atlas Biologicals, CO, USA) and 1% penicillin/streptomycin using gentleMACS Dissociator (Miltenyi Biotec, Bergisch Gladbach, Germany). After passing the homogenate through a 70 µm cell strainer (SPL Life Sciences, Gyeongi-do, Korea), cells were centrifuged at 314 x g at 4˚C for 7 minutes, followed by re-suspension and incubation in RBC lysis buffer (Biolegend, CA, USA) at 4˚C for 5 minutes. After washing (314 x g at 4˚C for 7 minutes), cells were kept in FACSFlow Sheath Fluid (BD Biosciences, CA, USA) containing 0.05% FBS (Atlas

Biologicals, CO, USA) on ice. For assessment of erythrocytes percentage, splenocytes after passing through cell strainer were processed directly with flow cytometry analysis steps without any red blood cell lysis. Bone marrow from femur and tibia were harvested from naïve and infected mice. Cell suspension was kept in FACSFlow Sheath Fluid containing 0.05% FBS on ice after passing through a 70 μm cell strainer.

To prepare for flow cytometry analysis, cells were incubated with Fc block (CD16/CD32 Fcγ III/II, Biolegend, CA, USA) (1/1000 dilution) at 4˚C for 30 minutes in the dark. Subsequently, $10^5$ cells per sample were incubate at 4˚C for 30 minutes in the dark with antibody cocktails specificity for different splenocytes populations, followed by flow cytometry analysis using BD Accuri C6 Plus and FACSCalibur flow cytometer (BD Biosciences, CA, USA). The total number of cells in each population was determined by multiplying the percentages of subsets within a series of marker negative or positive gates, with the total live cell number determined for each cell preparation in combination with microscopy live cell counts for every individual cell preparation. All gating strategies were included in S6 Fig.

**Antibodies and detection reagents.** The following antibodies (Biolegend, CA, USA) were added to 100 μl aliquots of 105 Fc-blocked splenocytes prepared as described above to make final 1/600 dilution: anti-CD1d-PE(clone 1B1), anti-CD23-APC (clone B3B4), anti-B220-FITC and anti-B220-PE-Cy7 (clone RA3-6B2), anti-CD93-APC and anti-CD93-PE-Cy7 (clone AA4.1), anti-CD138-PE-Cy7 (clone 281–2), anti-GL7-PE (clone GL7), anti-IgD-APC (clone 11-26c.2a), anti-Ter119-PE (clone TER-119), anti-Ly6G-Alexa488 (clone 1A8), anti-Ly6C-PE (clone HK 1.4). Anti-IgM-PE (clone X-54) was purchased from Miltenyi Biotec (Bergisch Gladbach, Germany) and diluted 1/11 in 100 μl aliquots of $10^5$ Fc-blocked splenocytes.

**Parasites isolation and VSG preparation.** Anticoagulated blood was collected from *T. evansi* Merzouga 93 infected mouse right before the first parasitemia peak. Parasites were separated from red blood cells using ion-exchange DEAE-cellulose (DE52) chromatography (Whatman, Maidstone, UK). Isolated parasites were collected in phosphate saline glucose buffer (44 mM NaCl, 57 mM Na2HPO4, 3 mM NaH2PO4, 55 mM glucose) and washed twice with DPBS (Invitrogen, Carlsbad, CA) by centrifuging at 1500 x g for 7 minutes. Soluble VSG was prepared as previous described [113]. In brief, isolated parasites were incubated in 1mL Baltz buffer (0.125 M phosphate buffer pH 5.5 containing 1% Glucose) on ice for 30 minutes, followed by 5 minutes incubation at 37˚C. The soluble VSG supernatant fraction was collected after centrifugation at 15000 x g for 10 minutes. Protein concentration was measured using the Bradford protein assay kit (Bio-rad, CA, USA).

**Quantification of anti-VSG and total antibody titers by ELISA.** Soluble VSG was coated in 96-well Half Area Clear Flat Bottom Polystyrene High Bind Microplate (Corning, NY, USA) at 4˚C overnight, at 0.2 μg/50 μl/well. Plasma was collected from infected mice on the same day as spleens were collected. A 1:2 serial dilution ranging from 1:100 to 1:204800 was prepared in DPBS (Invitrogen, Carlsbad, CA). Plasma Ab titer of IgM, IgG2b, IgG2c, and IgG3 were determined using horseradish peroxidase-labeled specific secondary antibodies (Southern Biotech, Alabama, USA). For detection of total plasma antibodies titers, 0.5 μg/100 μl/well of capture anti-mouse Ig antibody (Southern Biotech, Alabama, USA) was coated in 96 wells Immuno maxi-binding plate (SPL Life Sciences, Gyeonggi-do, Korea) at 4˚C overnight. Next, samples were processed as outlined above.

**Single cell RNA sequencing.** Spleen cells homogenates were prepared at three different time points (Fig 1C) as outlined above. After RBC lysis, washing and cell counting, approximately a total of 10,000 splenocytes from a single mouse per sample were loaded into single cell chips (10X Genomics) and partitioned using Gel Bead In-Emulsion (GEM) technology and a Chromium Controller (10X Genomics). Single cell libraries were prepared using the 10X Genomics Chromium Single Cell 3'reagent kit v3. The sequencing was conducted on a

NovaSeq6000 platform using TruSeq Illumina primers. The data matrix of each single cells was analyzed using the Cell Ranger Pipelines.

**Single-cell RNA-Seq data processing.** Raw sequences were processed using the Cell Ranger v4.0.0 pipeline developed by 10x Genomics. Briefly, the Illumina sequencer's base call files (BCLs) for each flow cell directory were demultiplexed into FASTQ files using the cellranger mkfastq command. The cellranger count command was used to perform alignment of reads against the mouse reference genome (mm10) using STAR v.2.5.1, which performs splicing-aware alignment of reads to the genome. After the alignment, a UMI matrix per gene per sample was generated. Feature-barcode matrices were generated individually for every sample (naïve, 14dpi, 42dpi).

For whole splenocytes visualization and analysis (S3 Fig), the output from 3 different time points were pooled using the cellranger aggr pipeline with normalization method set to "mapped", in which the read depths are normalized in terms of reads that are confidently mapped to the transcriptome. The Loupe Browser by 10x Genomics was used for gene expression-based clustering information for the cells, including t-SNE and UMAP projections and differential gene expression.

Alternatively, each samples' filtered feature-barcode matrix, which resulted from cellranger pipeline, was further cleaned by decontaminating cell free mRNA counts using SoupX package v1.5.0 [114] before individually loaded into Seurat package v3 [115] in R program v4.0.3 for quality control, clustering and downstream analysis. Cells that meet any of the following criteria were considered low quality and removed from the analysis: the number of features less than 200 or more than 4000 for naïve sample and 6000 for infected samples or the percentage of mitochondrial genes higher than 10%. Before normalization, genes *Gm42418* and *AY036118*, causing technical background noises [116] were removed from all three datasets. To mitigate the effects of cell cycle heterogeneity, the difference between the G2/M and S phase was regretted before dimensional reduction, clustering analysis, and visualization by Uniform Manifold Approximation and Projection (UMAP). Because our interests are B cell and PC populations, clusters expressing *Cd79a*, *Cd19* (B cells) or *Prdm1*, *Irf4* (PCs) from each sample separately were extracted using the 'subset' function and integrated to create one object containing B cells and PCs from 3 different time points (naïve, 14dpi, 42dpi). This object was submitted for another round of dimensional reduction and clustering analysis, resulting in 13 clusters which were then annotated to 6 B cells/PCs populations. The final annotated object was visualized using BioTuring Browser (version 2.7.34 for Mac OS, developed by BioTuring Inc., San Diego California USA). Differential expression analysis between 2 populations was performed by the Wilcoxon Rank Sum Test and visualized by the R package EnhancedVolcano [117]. Top 100 differential expressed genes (ranked on fold-change with p-value $\leq$ 0.05 after filtering ribosomal genes) were used for gene ontology analysis performed by R package clusterProfiler [118].

**Immunohistochemistry and H&E staining of frozen sections.** Spleens from mice were collected, fixed using Formalin solution neutral buffered 10% (Sigma-Aldrich, Missouri, USA), and soaked in 30% sucrose before being embedded in FSC 22 Frozen Section Media (Leica Biosystems, IL, USA) followed by snap freezing with dry ice. Tissue sectioning was done by the Laboratory Animal Research Facility in National Cancer Center, Korea. 6 µm-thick-sections were fixed for 15 minutes in acetone and methanol with a 1:1 ratio before blocked in PBS containing 10% FBS (Atlas Biologicals, CO, USA) and 0.3% TWEEN 20 (Sigma-Aldrich, Missouri, USA) for 1 hour at room temperature. Staining with mixture of 5 µg /ml FITC conjugated anti-B220 (clone RA3-6B2, Biolegend, CA, USA) and 20 µg/ml biotinylated anti-AID (clone mAID-2, eBioscience, CA, USA) was carried out at 4oC, overnight. Sections were washed and incubated with 1/500 dilution of Cyanine3 Streptavidin (Biolegend, CA, USA) at

room temperature for 1 hour before counterstaining with DAPI. Sections were mounted in VECTASHIELD Antifade mounting medium (Vector Laboratories, CA, USA) and viewed with Olympus iX83 microscope. For H&E staining, fixed sections were stained using the H&E staining kit (Abcam, Cambridge, United Kingdom) following the manufacture's instruction.

**Passive plasma transfer.** Plasma from WT and AID$^{-/-}$ 7dpi infected mice was passively transferred into 3dpi infected C57BL6/N WT mice at 100 μL per mouse using the intravenous (IV) route of administration. Parasitemia at different time points were examined as outlined above.

***In vitro* trypanolytic assay.** *T. evansi* Merzouga 93 parasites were isolated as mentioned above. $10^6$ parasites per sample were cultured in HMI9 [119] supplemented with 10% plasma from AID$^{-/-}$/WT mice or culture media as control for 4 hour in 37˚C. After incubation, live parasites in each sample were quantified by mixing parasite with Trypan blue (1:1 ratio) and counting under light microscope.

## Quantification and statistical analysis

GraphPad Prism v.8.3 (GraphPad Software Inc. San Diego, CA) was used for final data presentation and sktatistical result analysis. Unless otherwise stated, data were compared with naïve using student's t-test. Means are given as + standard deviation (SD). Significance was determined as follows: $^*p \leq 0.05$; $^{**}p \leq 0.01$, and $^{***}p \leq 0.001$ or where indicated, as ns $\geq 0.05$.

## Supporting information

**S1 Fig. Infection-induce antibody response. (A)** Total IgM, IgG1, IgG2b, IgG2c and IgG3 isotype serum antibody titers during *T. evansi* infection. **(B)** Anti-VSG IgM, IgG1, IgG2b, IgG2c and IgG3 isotype serum antibody titers during *T. evansi* infection. Data is presented as log2 fold change of OD50 values for total antibody titers and End-point titer values for Anti-VSG antibody titers in comparison to Naïve samples. Data is presented as mean + SD of 3 individual samples, from one of three independent experiments, with significant differences compared to naïve control samples ns: non-significant by Student's t-test.
(TIF)

**S2 Fig. Infection-induced splenomegaly, erythro/granulopoiesis and B cells homeostasis loss. (A)** Splenomegaly and splenocytes number at different time points during *T. evansi* infection. **(B)** Percentage of B lymphocytes, Granulocytes and Erythrocytes in splenocytes during *T. evansi* infection. **(C)** Percentage of B cell populations during *T. evansi* infection. All data is represented as the mean + SD of three individual samples, in one of three representative experiments, with significant differences compared to naïve control mice * p≤0.05; ** p≤0.01; *** p≤ 0.001by Student's t-test.
(TIF)

**S3 Fig. ScRNA sequencing analysis of splenocytes at 3 different time points. (A)** UMAP projection of splenocytes from 3 time points colored by graph-based clusters. **(B)** Log$_2$ expression of classical markers for B cells (blue boxes) and Plasma cells (Red boxes) across graph-based clusters. **(C)** UMAP projection of splenocytes colored based on expression of *Cr2* (top), *Fcer2a* (middle), *Ighd* (bottom).
(TIF)

**S4 Fig. Heatmap composed by combing the 10 highest differential expressed genes for each splenic B cell population and plasma cells.**
(TIF)

**S5 Fig. Alterations in B cell and PC populations of AID$^{-/-}$ and WT infected mice during *T. evansi* infection. (A)** One Representative profile of 9 individual measurements by flow cytometry analysis of MZB, FoB, PCs of WT (upper left) and AID$^{-/-}$ (upper right) naive mice. Percentage values of MZB, FoB and PCs of WT and AID$^{-/-}$ naïve mice (lower panel).) Data is presented as mean + SD, with significant differences compared to naïve WT mice by Student's t-test. ns: non-significant. **(B)** Flow cytometry analysis of Early B lineage cells in bone marrow of WT (top) and AID$^{-/-}$ (bottom) infected mice. One representative data set of 9 measurements is shown for each group.
(TIF)

**S6 Fig. Gating strategies for flow cytometry analysis. (A-B)** Gating strategy to assess cell number and percentage of MZB cells, FoB cells, PCs and GC-like B cells populations. **(C)** Gating strategy to measure expression of CD23 on FoB cells. **(D)** Gating strategy to visualize CD93$^+$ early B cell lineage in bone marrow. **(E-F)** Gating strategy to assess percentage of granulocytes and erythrocytes.
(TIF)

**S1 File. Full list of differentially expressed genes resulted from DEG analysis of 14dpi derived MZB/FoB cells compared with naïve derived cells.**
(XLSX)

**S1 Table. Reagents and resources information.**
(DOCX)

## Acknowledgments

We would like to express our gratitude towards Dr. Joar Esteban Pinto Torres for his support in making our graphical abstract. We also thank members of the Laboratory Animal Research Facility at National Cancer Center, Korea for their technical support related to tissue processing for immunohistochemistry.

## Author Contributions

**Conceptualization:** Hang Thi Thu Nguyen, Stefan Magez, Magdalena Radwanska.

**Data curation:** Hang Thi Thu Nguyen, Stefan Magez, Magdalena Radwanska.

**Formal analysis:** Hang Thi Thu Nguyen, Stefan Magez, Magdalena Radwanska.

**Funding acquisition:** Stefan Magez, Magdalena Radwanska.

**Investigation:** Hang Thi Thu Nguyen, Stefan Magez, Magdalena Radwanska.

**Methodology:** Hang Thi Thu Nguyen, Stefan Magez, Magdalena Radwanska.

**Project administration:** Stefan Magez, Magdalena Radwanska.

**Resources:** Stefan Magez, Magdalena Radwanska.

**Software:** Hang Thi Thu Nguyen, Robin B. Guevarra.

**Supervision:** Stefan Magez, Magdalena Radwanska.

**Validation:** Hang Thi Thu Nguyen, Stefan Magez, Magdalena Radwanska.

**Visualization:** Hang Thi Thu Nguyen.

**Writing – original draft:** Hang Thi Thu Nguyen, Robin B. Guevarra, Stefan Magez, Magdalena Radwanska.

**Writing – review & editing:** Hang Thi Thu Nguyen, Stefan Magez, Magdalena Radwanska.

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
