## [Decision Letter · Decision Letter 0]

30 Apr 2021

Dear Prof. Magez,

Thank you very much for submitting your manuscript "Single-cell transcriptome profiling and the use of AID deficient mice reveal that B cell activation combined with antibody class switch recombination and somatic hypermutation undermine the control of experimental trypanosomosis." for consideration at PLOS Pathogens. As with all papers reviewed by the journal, your manuscript was reviewed by members of the editorial board and by several independent reviewers. In light of the reviews (below this email), we would like to invite the resubmission of a significantly-revised version that takes into account the reviewers' comments.

Reviewer #2 has raised some important issues that need to be addressed. In particular, a mechanistic data set supporting the functional importance of natural IgM should be provided in order to support the conclusion that IgM is more effective than IgG. Also, please provide the survival data (other than only parasitemia) comparing infected AID-/- and WT animals.

We cannot make any decision about publication until we have seen the revised manuscript and your response to the reviewers' comments. Your revised manuscript is also likely to be sent to reviewers for further evaluation.

Sincerely,

Jude Ezeh Uzonna, Ph.D., DVM

Guest Editor

PLOS Pathogens

David Sacks

Section Editor

PLOS Pathogens

Kasturi Haldar

Editor-in-Chief

PLOS Pathogens

orcid.org/0000-0001-5065-158X

Michael Malim

Editor-in-Chief

PLOS Pathogens

orcid.org/0000-0002-7699-2064

Reviewer's Responses to Questions

**Part I - Summary**

Reviewer #1: This group has previously shown that infection of Tryanosoma brucei induces cell death of splenic B cells and disrupts vaccine-induced memory responses. In the current study, with the use of multiple approaches including flow cytometry and single cell RNA-sequencing analysis, they characterized the antibody and B cell responses in a mouse model of infection with T. evansi. They found that the parasites induce activation of splenic IgM+CD1d+ marginal zone and IgMIntIgD+ follicular B cells, associated with enhanced plasma levels of IgG2c Abs. They demonstrated that GC-like B cell induction occurs during infection which is independent of GC formation and that ablation of the bone marrow immature B cell compartment prevented mature peripheral B cell replenishment. Finally, they showed that infected AID-/- mice lacking anti-trypanosome IgGs exhibited a better clearance of the parasites compared to infected WT mice with the major contribution of IgM Abs. Overall, this is an insightful manuscript, providing novel insights into our understanding how the parasites escape the host immune responses. I have some minor comments for consideration.

1) In the introduction (line 78-82), the authors mentioned several mechanisms of avoiding killing by the host immune responses including complement. However, recent data demonstrate that complement activation play an essential role in intravascular clearance of trypanosomes.

2) In the results, Fig. 6, they showed that infected AID-/- mice displayed significantly lower parasitemia. Do infected AID-/- mice survive significantly longer than infected WT mice?

3) In the discussion (line 417-420), they discussed importance of IgM Abs in controlling parasitemia. Consistent with the current study, the authors have previously shown that IgM Abs are required to control the prarasitemia during infection with T. evansi (Baral et. al., JID 2007 195(10):1513-20; PMID: 17436232), whereas IgM Abs paly a limited role during T. brucei infection (Magez et. al. PLoS Pathog. 2008, 4(8):e1000122; PMID: 18688274). Recent publication demonstrated that IgM Abs play a prominent role in activation of complement for intravascular clearance of T. congolense (Liu et. al, 2019 PNAS, 116(48):24214-24220; PMID: 31723045). As one important part of this manuscript is related to the function of IgM Abs, the author should expand the discussion by comparing the role of IgM Abs among those different trypanosome species. IgM Abs activate complement more efficiently than IgG Abs, given that complement is critically involved in intravascular clearance of trypanosomes. This point should be also considered when comparing IgM and IgG Abs. Finally, in line 419-420: “IgMs are crucial for trypanosome phagocytosis by CR3+ macrophages (100)”. This is true for in vitro phagocytosis by purified peritoneal macrophages (ref 100). However, recent in vivo data have shown that complement receptor CRIg but not CR3 expressed by Kupffer cells plays an essential role in intravascular clearance of T. congolense parasites (Liu et. al, 2019 PNAS, 116(48):24214-24220: PMID: 31723045). This knowledge must be updated in the manuscript. Indeed, Kupffer cells express abundant CRIg but limited CR3 in vivo (PMID: 16530040).

Reviewer #2: This study re-examines a number of issues regarding the role of B cells and antibodies in Trypannosome infection using higher resolution approaches such as single cell RNA sequencing. The authors further provide functional evidence for the role of natural/innate IgM antibody in controlling parasite growth. Their conclusions are that:

1) natural IgM is very important for controlling parasite growth,

2) the “B cell depletion” observed upon infection includes strong reduction in early/transitional B cells (including in bone marrow)

3) germinal centre-like B cells are generated, peaking around day 14, even though proper germinal structures don’t appear to be present in spleen.

4) innate-like marginal zone and B1 populations are most strongly depleted, by conversion to short lived effector memory/plasma cells,

5) depletion of follicular B cells is not as complete, and previous studies may have over-estimated this due to CD23 down regulation

6) IgG responses (particularly IgG2c) are less effective against the parasites than IgM, and may represent an evasion mechanism

Reviewer #3: This is an important manuscript and deserves publication.

There is a long history of work on approaches to a vaccine for trypanosomiasis. Often the work on the pathogen has become divorced form the immunology of the host response. This type of manuscript is well grounded in looking at a realistic animal model (no such models are perfect and some have been shown to be misleading, but this is reasonable) and in using a non-clonal population of a recent field isolate of the parasite.

The approaches used are complementary and produce an interesting result. Importantly, this result may well have general applicability in other pathogen infections such as malaria. The result – that trypanosomes drive an early-stage IgMs/IgG2c switch and terminal PC differentiation followed by reduction of B-cell compartment (little replenishment) – provides a now way of thinking about this infection.

This concept trypanosomes drive immune maturation to evade high avidity IgM interactions and hence subvert the immune response is novel and raises an important concept. Moreover, it is important that the parasite vaccine community is aware of this since focussing on only certain parameters such as general antibody levels etc may well not be good indicators of vaccine effectiveness in the field. Also, it provides a working hypothesis of why bacterial vaccines might fail in tryp infected animals.

This data along with the hypotheses outlined are important and will greatly assist progress of the field. They provide a new set of ideas.

**Part II – Major Issues: Key Experiments Required for Acceptance**

Reviewer #1: (No Response)

Reviewer #2: This is a very interesting study that clarifies several important aspects, however some conclusions are based on relatively little data and some improvements in data and writing are needed. As the data are mostly observational, the experiments demonstrating functional mechanisms should be strengthened.

1) The data supporting the functional importance of natural IgM in vitro is limited. To support the conclusion that IgM is more effective than IgG, isolated IgM and IgG would need to be compared. Naïve/post-infection serum should be compared. Is AID plasma more effective due to different quantity or quality of IgM Ab or other reasons?

2) The data supporting the functional importance of natural IgM in vivo is limited. Plasma transfer experiments should compare wild type and AID KO naïve/post-infection serum.

3) Line 272/73 suggest that IgG2c is “detrimental”. Implying that IgG2c is somehow detrimental to the immune response does not seem justified. The data address whether expanded natural IgM is beneficial, but there is not data addressing whether IgG2c is detrimental. To support line 289, where is the evidence that IgG2c is a “decoy mechanism”? This conclusion is repeated several times, and in lines 435 and 460 another mechanism “competition for antigen” is suggested (even though the authors find that the “vast majority of plasma cells are IgM+” on line 385). As there is no evidence in the manuscript for any of these negative effects of IgG2c, these should not be stated as conclusions but as possibilities for future investigation. For this reviewer, the important challenge to focus on is how to boost or maintain IgM responses prior to/during infection, not how to suppress so called detrimental IgG responses.

Reviewer #3: Overall I have no major points here. The authors use a number of different approaches ...cell analysis, population analysis and markers, plasma transfer, knockout mice to reach a holistic conclusion. I am not an immunologisy=t but in this type of work I think one needs to focus on the holistic rather than always asking for more detail.

**Part III – Minor Issues: Editorial and Data Presentation Modifications**

Reviewer #1: (No Response)

Reviewer #2: 1) Line 123. Instead of arbitrarily mentioning day 42, describe the survival curve showing when the mice start to die

2) Line 127/28 –5-10 fold increases in IgG isotypes after infection cannot be described as low, just less than IgG2c. To justify the focus of IgG2c, the data comparing anti-VSG titres for different isotypes should be shown.

3) Starting with Fig 1, complete flow cytometry gating strategies need to be illustrated for all cell populations reported.

4) Line 131 – “decline in cell numbers” Description of flow cytometry data needs to consistently refer to altered cell frequencies, unless referring to absolute cell number data. Its not acceptable to bury absolute cell numbers in supplementary data and not even mention it. I think this should be included in the primary figure and explained clearly as its critical for the reader to understand whats going on.

5) The absolute number data include erythrocytes but method for spleen cell preparation, counting and flow cytometry gating for this analysis are not explained.

6) Line 141 – what is “graph-based clustering? Phenograph algorithm?

7) Line 176 – this concluding statement doesn’t make sense as flow cytometry data for several populations have not been shown at this point

8) Line 212 - is a relationship between transitional and B1 cell depletion being implied here?

9) Line 225/26 - this is a strange way to describe GC B cell expansion – for consistency it would be better to express the result as percentages of total B lineage as in Fig1

10) Line 360 “short-lived atMBCs” – what is the evidence they are short-lived?

11) Line 379 – I don’t think histological evidence of absence of GC has been clearly shown. The immunofluorescence images shown do not demonstrate this. Staining such as GL7/IgD/AID would be needed, and comparing infected mice spleen with positive control samples containing GC structures would be needed.

12) Line 405 - how can AID deficiency affect response to IgG-inducing signals? More accurate to say AID-deficient cells can’t perform class switch recombination

13) Is the single cell RNAseq data from a single mouse per time point or pooled cells from multiple mice?

14) Fig 3D - are the gene functional groups shown the most statistically significant ones, or selected based on other criteria?

15) Fig 4E and F aren't related to A-D parts, thus its not clear why these should be combined. I felt unclear on the takeaway conclusions regarding B1 cells – unlike MZ cells it seems they are expanded and activated rather than reduced at day 14?. This reviewer was not familiar with the markers used to identify the B1 cell population. It would be helpful to compare with flow cytometry analysis of B1 cells as done for all of the other populations.

16) Fig 7 should show baseline differences in AID -/-

17) In several places the figure legends do not indicate the number of mice. BOTH the number of mice in the experiment shown and the number of repeat experiments should be indicated: Figure 1C (# repeat exp), 1E (how many mice per time point? How many repeats), Fig 2 (how many mice), Fig 4D (# of mice and repeats), Fig 7B (how many repeats), S5B (# of mice, repeats)

18) Numerous typos and wrong words were noted throughout and need to be corrected.

Reviewer #3: POINTS:

The paper will be of interest to immunologists, parasite immunologists and those interested in the parasite. For the latter the complexities of immunology language use could be clarified in places. For instance in places IgM subtypes are defined but in other places the term “natural IgM” is used. I understand the general meaning of natural IgM and maybe it is just me - but I had a little difficulty understanding which population of IgM was being referred to ..for instance on page 10…the terms Natural IgM, VSG binding IgM, IgMs are used.

There are a number of English usage / typos that would be usefully amended before publication: for instance particularly in the discussion –

Line 332 treat?

Line 360. lallter?

Line 400 requires result ?

Line 417 require to speculate?

Line 427 VSG a homodimer?

Line 431 pulse-chance?

PLOS authors have the option to publish the peer review history of their article (what does this mean?). If published, this will include your full peer review and any attached files.

Reviewer #1: No

Reviewer #2: **Yes: **Aaron Marshall

Reviewer #3: No
---

## [Decision Letter · Decision Letter 1]

11 Oct 2021

Dear Dr. Magez,

We are pleased to inform you that your manuscript 'Single-cell transcriptome profiling and the use of AID deficient mice reveal that B cell activation combined with antibody class switch recombination and somatic hypermutation do not benefit the control of experimental trypanosomosis' has been provisionally accepted for publication in PLOS Pathogens.

Best regards,

Jude Ezeh Uzonna, Ph.D., DVM

Guest Editor

PLOS Pathogens

David Sacks

Section Editor

PLOS Pathogens

Kasturi Haldar

Editor-in-Chief

PLOS Pathogens

orcid.org/0000-0001-5065-158X

Michael Malim

Editor-in-Chief

PLOS Pathogens

orcid.org/0000-0002-7699-2064

Reviewer Comments (if any, and for reference):

Reviewer's Responses to Questions

**Part I - Summary**

Reviewer #1: The authors have appropriately addressed my comments.

Reviewer #2: The authors have undertaken a thorough revision addressing most of the issues identified. One minor exception was the plasma transfer experiments where I felt that it would be informative to also examine pre-infection plasma to determine whether elevated natural antibodies in AID-/- might be sufficient to provide partial protection. The in vitro trypannocidal activity of AID -/- plasma suggests this could be the case.

**Part II – Major Issues: Key Experiments Required for Acceptance**

Reviewer #1: (No Response)

Reviewer #2: (No Response)

**Part III – Minor Issues: Editorial and Data Presentation Modifications**

Reviewer #1: (No Response)

Reviewer #2: (No Response)

PLOS authors have the option to publish the peer review history of their article (what does this mean?). If published, this will include your full peer review and any attached files.

Reviewer #1: No

Reviewer #2: **Yes: **Aaron Marshall

---

## [Editor Report · Acceptance letter]

8 Nov 2021

Dear Dr. Magez,

We are delighted to inform you that your manuscript, "Single-cell transcriptome profiling and the use of AID deficient mice reveal that B cell activation combined with antibody class switch recombination and somatic hypermutation do not benefit the control of experimental trypanosomosis," has been formally accepted for publication in PLOS Pathogens.

Best regards,

Kasturi Haldar

Editor-in-Chief

PLOS Pathogens

orcid.org/0000-0001-5065-158X

Michael Malim

Editor-in-Chief

PLOS Pathogens

orcid.org/0000-0002-7699-2064